A heat-shock 20 protein isolated from watermelon (ClHSP22.8) negatively regulates the response of Arabidopsis to salt stress via multiple signaling pathways

He Yanjun 1
Yao Yixiu 1 2
Li Lili 1 2
Li Yulin 1 2
Gao Jie 2
Fan Min 1 fanm@zaas.ac.cn
1 Zhejiang Academy of Agricultural Sciences, Institute of Vegetables , Hangzhou, Zhejiang , China
2 College of Forestry and Horticulture, Xinjiang Agricultural University , Urumqi, Xinjiang , China
Gagaoua Mohammed
Electronic publication date: 2021 Mar 1
Publication date: 2021
Volume: 9
Electronic Location ID: e10524
Received 2020 Jul 22; Accepted 2020 Nov 17
Copyright: © 2021 He et al.
Copyright year: 2021
Copyright holder: He et al.
License: This is an open access article distributed under the terms of the Creative Commons Attribution License, which permits unrestricted use, distribution, reproduction and adaptation in any medium and for any purpose provided that it is properly attributed. For attribution, the original author(s), title, publication source (PeerJ) and either DOI or URL of the article must be cited.
License URL: https://creativecommons.org/licenses/by/4.0/

Keywords: ClHSP22.8, Watermelon (Citrullus lanatus), Salt stress, ABA, Arabidopsis

Funding: National Key Research & Development Program of China 2018YFD0100703 Natural Science Foundation of Zhejiang Province LQ18C150003 National Natural Science Foundation of China 31772332 This research was funded by the National Key Research & Development Program of China (2018YFD0100703), Natural Science Foundation of Zhejiang Province (LQ18C150003), and National Natural Science Foundation of China (31772332). The funders had no role in study design, data collection and analysis, decision to publish, or preparation of the manuscript.

==============================
Heat-shock protein 20s (HSP20) were initially shown to play a role during heat shock stress; however, recent data indicated that HSP20 proteins are also involved in abiotic stress in plants. Watermelon is known to be vulnerable to various stressors; however, HSP20 proteins have yet to be investigated and characterized in the watermelon. In a previous study, we identified a negative regulator of salt stress response from watermelon: ClHSP22.8, a member of the HSP20 family. Quantitative real-time PCR (qRT-PCR) and promoter::β-glucuronidase (GUS) analysis revealed that ClHSP22.8 was expressed widely in a range of different tissues from the watermelon, but particularly in the roots of 7-day-old seedlings and flowers. Furthermore, qRT-PCR and GUS staining showed that the expression of ClHSP22.8 was significantly repressed by exogenous abscisic acid (ABA) and salt stress. The over-expression of ClHSP22.8 in Arabidopsis lines resulted in hypersensitivity to ABA and reduced tolerance to salt stress. Furthermore, the expression patterns of key regulators associated with ABA-dependent and independent pathways, and other stress-responsive signaling pathways, were also repressed in transgenic lines that over-expressed ClHSP22.8. These results indicated that ClHSP22.8 is a negative regulator in plant response to salt stress and occurs via ABA-dependent and independent, and other stress-responsive signaling pathways.

Introduction

Heat-shock proteins (HSPs) act as molecular chaperones and are found in all species of plants. HSPs help to protect their target proteins from denaturation, misfolding, and aggregation, during times of stress (Papsdorf & Richter, 2014; Asea, Kaur & Calderwood, 2016). Previous research has shown that HSPs can be classified into six groups based on molecular weight: HSP100, HSP90, HSP70, HSP60, HSP20 (or small heat-shock protein) and ubiquitin (Asea, Kaur & Calderwood, 2016; Sun & MacRae, 2005). HSP20 is now known to be the largest and best studied family of the HSP families (Basha, O’Neill & Vierling, 2012). Numerous studies have revealed specific roles for plant HSP20 proteins in a range of abiotic stress responses. For example, transgenic plants that over-expressed Heat-shock protein 20s (HSP20s) was shown to exhibit enhanced tolerance to heat, including Arabidopsis, rice, wheat, maize and Chenopodium (Sun & MacRae, 2005; Sun et al., 2012; Murakami et al., 2004; Charng et al., 2006; Shakeel, Heckathorn & Luthe, 2012; Khurana, Chauhan & Khurana, 2013). Another study showed that the over-expression of OsHSP20s in rice led to enhanced tolerance to stress caused by ultraviolet-B radiation, salt, drought, and dehydration (Murakami et al., 2004; Zou et al., 2012; Kaur et al., 2015). Other research studies have shown that the over-expression of OsHSP17.0 or OsHSP23.7 led to an improvement in the drought and salt tolerance of rice and that this involved a reduction of membrane damage and increased expression of protective molecules (Zou et al., 2012). Research has shown that OsHSP18.2 is implicated in seed vigor and longevity and can improve germination and the successful creation of seedlings under abiotic stress (Kaur et al., 2015). Promoter analyses further revealed that the over-expression of TaHSP26 in wheat could be induced by heat, cold, salt and drought (Chauhan et al., 2012). Furthermore, the over-expression of TaHSP23.9 in wheat led to an enhancement in the tolerance to heat and salt stresses (Wang et al., 2020). Another study, involving the ectopic expression of LimHSP16.45 led to an increase in the activities of superoxide dismutase (SOD) and catalase (CAT), thus improving the vigor of seed germination in Arabidopsis under salt stress (Mu et al., 2013). The over-expression of ZmHsp16.9 in tobacco led to an enhancement in the activities of peroxidase (POD), CAT and SOD, and an increase in oxidative stress tolerance (Sun et al., 2012). HSP20s have been found to regulate the plant response to salt stress via abscisic acid (ABA) signaling pathways. In Capsicum annuum, CaHsp22.5 was shown to modulate plant ABA signaling and participate in response to salt stress (Li et al., 2018). Other research, carried out in creeping bentgrass, showed that AsHSP17 and AsHSP26.8a mediate ABA-dependent and independent and other stress signaling pathways to negatively regulate plant responses to salt stress (Sun et al., 2016, 2020).

Watermelon (Citrullus lanatus L.) is an economically important cucurbit crop that is cultivated across the world. However, it is vulnerable to a variety of adverse environmental conditions (Guo et al., 2013). As one of the most important stressors, salinity stress can lead to serious limitations in the yield and quality of watermelon (Yetr & Uygur, 2009). HSP20s are the most abundant HSP sub-type in plants and are known to play important functions in a variety of stress responses (Sun et al., 2012, 2016, 2020; Zou et al., 2012; Wang et al., 2020; Mu et al., 2013; Li et al., 2018). However, we know very little about the specific role of watermelon HSP20s with regards to abiotic stress tolerance.

In a previous study, we identified the HSP20 gene family in watermelon and analyzed their expression patterns in response to different stresses (He et al., 2019). In the present study, we characterized an HSP20 gene (ClHSP22.8) from the watermelon. Quantitative real-time PCR (qRT-PCR) and promoter::β-glucuronidase (GUS) assays showed that the expression of ClHSP22.8 was repressed by exogenous ABA and salt treatment. Next, we successfully constructed Arabidopsis lines that over-expressed ClHSP22.8 and used these lines to investigate their sensitivity to ABA and salt tolerance. We analyzed the expression profiles of genes related to ABA and the stress response of plants that over-expressed ClHSP22.8 under salt treatment. Functional studies of ClHSP22.8 will not only provide a better understanding of the specific roles of HSP20s in the adaption of watermelon to salt stress but may also provide insight into the potential signaling processes in response to stressful conditions.

Materials and Methods

Identification of ClHSP22.8 and bioinformatics analysis

First, we downloaded the amino acid sequence of ClHSP22.8 from the Cucurbit Genomics Database (accession number: Cla017945). We then used ExPASy (http://web.expasy.org/computepi/) to calculate the molecular weight (MW) and isoelectric point (pI). Conserved domains in the ClHSP22.8 protein were confirmed using the SMART database (http://smart.emblheidelberg.de/). Next, we downloaded the protein sequences of various ClHSP22.8 orthologs in Arabidopsis, rice, tomato, soybean, switchgrass, and cucumber, from Phytozome (http://phytozome.jgi.doe.gov/pz/portal.html). Then, we performed phylogenetic analysis, based on the full-length protein sequences, using the MEGA 5.0 program and the neighbor-joining (NJ) method with 1,000 bootstrap replicates (Tamura et al., 2011). Multiple sequence alignment of the predicted peptide sequences of the conserved-crystallin (ACD) domain was then carried out using Clustal X version 1.81 with default parameters (Thompson et al., 1997). The ClHSP22.8 promoter sequence was also downloaded from the Cucurbit Genomics Database and subsequently analyzed via PlantCARE (http://bioinformatics.psb.ugent.be/webtools/plantcare/html/).

Plant materials and growth conditions

We used the watermelon advanced inbred line “JJZ-M” for all expression analyses. These plants were grown in a growth chamber in temperature-controlled greenhouses under day/night temperatures of 28/22±1 °C, a light intensity of 200 μmol m−2 s−1, and a 16-h light/8-h dark photoperiod. Three-week-old watermelon seedlings were used for treatments involving exogenous ABA and salt stress treatments; these treatments involved the seedlings being sprayed with 100 μM of ABA and 200 mM of NaCl, respectively (He et al., 2019). The second true leaf on each plant was sampled at time 0 (control) and then again at 1, 4 and 12 h after treatment. Arabidopsis thaliana ecotype “Columbia” (wild type, WT) plants were used for the construction of transgenic plants. These plants were kept at 24/22 °C (16 h-day/8 h-night) with 65% relative humidity to yield transgenic lines. Both transgenic and WT plants were cultivated under the same growth conditions.

Quantitative real-time PCR

Total RNA was isolated from samples of both watermelon and Arabidopsis. Reverse transcription was then performed using the PrimeScript RT reagent kit (Takara, China) in accordance with the manufacturer’s instructions; for each sample, approximately 1 μg of total RNA was reverse transcribed into cDNA. qRT-PCR reactions were performed on a CFX96 Real Time PCR System (Bio-Rad, USA) using the following cycle conditions: 30 s at 95 °C; followed by 40 cycles of 5 s at 95 °C, and 45 s at 55 °C; this was followed by 1 cycle of 1 min at 95 °C, 30 s at 50 °C and 30 s at 95 °C. Two biological and three technical replicates were carried out for each sample; these reactions involved a reaction volume of 15 μL and the SYBR Premix Ex Taq kit (Toyobo, Japan). We used the watermelon β-actin gene and the Arabidopsis ACTIN2 gene as reference sequences for primer design (Table S1) and relative gene expression was calculated using the 2−ΔΔCt method.

Promoter assay by GUS histochemical staining

In order to investigate the tissue-specific expression of ClHSP22.8, we amplified a 1601 bp upstream promoter sequence using specific primer pairs (Table S1). We then ligated this fragment with the pBI101 plasmid vector that could be subsequently transformed into Arabidopsis. The reporter construct containing the GUS reporter gene driven by the ClHSP22.8 promoter region was named ProClHSP22.8::GUS (Fig. S1). Transgenic Arabidopsis seedlings were obtained using the floral dip method (Xiuren et al., 2006). Transgenic Arabidopsis seeds were screened using 100 mg L−1 kanamycin (KanR). Positive transgenic plants were identified by GUS histochemical staining. T3 transgenic lines were also screened and harvested for further phenotypic observation.

The 7-day-old Arabidopsis transgenic seedlings created from the T3-generation grown on 1/2 MS medium were transferred to 1/2 MS medium with and without 100 μM of ABA and 200 mM of NaCl. After 24 h, the transgenic seedlings were GUS stained using a GUS Histochemical Staining Kit (O’BioLab, Beijing, China) in accordance with the manufacturer’s guidelines. After removing chlorophyll with 70% ethanol, we analyzed the seedlings and acquired typical digital images using a stereomicroscope (STEMI SV11, Zeiss, Jena, Germany).

Plasmid construction and generation of transgenic plants

The coding sequence (CDS) of ClHSP22.8 was amplified using a pair of specific primers: ClHSP22.8-F and ClHSP22.8-R (Table S1). The amplicons produced by PCR were subsequently digested and ligated into the pBI121 vector (Fig. S1). Subsequently, the pBI121-p35S::ClHSP22.8 vector was transformed into Arabidopsis using the floral dip method. Transgenic Arabidopsis lines were confirmed by PCR using two specific primers: HSP22.8-S and GUS-A (Fig. S2). The progenies of these plants were screened using 100 mg L−1 KanR, as described earlier. As a result, four independent homozygous transgenic lines were created; we named these OE22.8-1, OE22.8-2, OE22.8-3 and OE22.8-4. T3 transgenic lines were screened and harvested for further phenotypic observation.

ABA and salt tolerance in transgenic Arabidopsis lines

The T3-generation transgenic lines (OE22.8-1–OE22.8-4) were grown with WT seedlings on 1/2 MS medium with or without 100 μM of ABA and 200 mM of NaCl, respectively. After 7 days, we photographed these plants so that we had a record of their relative phenotypes. The growth status and root length of two-week-old plants were measured, and samples were taken for qRT-PCR. Three independent biological replicates were analyzed; each replicate involved over 30 seedlings.

Statistical analysis

Data were analyzed by a two-tailed Student’s t-test or by one-way analysis of variance (ANOVA) using SPSS version 18.0 (IBM, Chicago, IL, USA). *P < 0.05 and **P < 0.01 were considered to be significant and highly significant, respectively.

Results

Isolation and bioinformatics analysis of ClHSP22.8

The length of the full-length coding sequence (CDS) for the ClHSP22.8 gene was 582 bp and coded for a protein containing 193 amino acids. The isoelectric point (pI) of the protein was 7.76, and the molecular weight (MW) was 22.85 kDa (Table S1). ClHSP22.8 on Chr 10 was mapped to a segmentally duplicated region that was shared with ClHSP16, ClHSP17.6C and ClHSP17.6D, respectively (Table S1). The ClHSP22.8 protein shared a conserved α-crystallin ACD/HSP20 domain (thus showing conservation of region I and II) with its orthologs (Fig. 1A). Phylogenetic analysis further showed that ClHSP22.8 and CsHSP23.7 in the cucumber formed a separate branch that was distant from a range of other orthologs from Arabidopsis, rice, tomato, soybean and switchgrass (Fig. 1B).

Figure 1 Phylogenetic and amino acid sequence analysis of ClHSP22.8 orthologs in various species.

(A) Amino acid sequence alignment of α-crystallin ACD/HSP20 domain from Arabidopsis (At), rice (Os), tomato (Sl), soybean (Gm), switchgrass (Pv), cucumber (Cs) and watermelon (Cl). Conserved region I and II were indicated by red boxes. (B) Phylogenetic analysis of ClHSP22.8 protein orthologs from these orthologs. Phylogenetic analysis based on full-length protein sequences was performed using the MEGA 5.0 program by the neighbor-joining (NJ) method with 1,000 bootstrap replicates.

Spatial and temporal expression of ClHSP22.8 in watermelon

Next, we determined the spatial and temporal expression profiles of ClHSP22.8 via qRT-PCR. Data indicated that ClHSP22.8 was widely expressed across a range of different tissues in the watermelon and showed the lowest level in the fruit; higher levels of expression were observed in female flowers and roots (51.37- and 26.63-fold compared to that in the fruit) (Fig. 2A). In order to further characterize the tissue-specific expression of the ClHSP22.8 gene, we amplified a 1,601 bp fragment from a region that was upstream of the ATG start codon in the ClHSP22.8 gene. We then transfected this fragment into Arabidopsis in order to drive β-glucuronidase (GUS) gene expression (Fig. S1; Figs. 2B–2G). Results from GUS histochemical staining indicated that the GUS protein was expressed at the highest levels in roots; the next highest level of expression was seen in the cotyledons, especially in the leaf vein. Signals were also detected in the vascular tissue of hypocotyls in 7-day-old Arabidopsis seedlings (Fig. 2B). In adult Arabidopsis plants, the GUS signal was expressed at the highest levels in flowers, followed by leaves. Only weak signals were detected in the stem, silique and roots (Figs. 2C–2G).

Figure 2 Spatial and temporal expression patterns of ClHSP22.8.

(A) ClHSP22.8 expression levels in root (R), stem (S), leaf (L), female flower (Ff), male flower (Mf) and fruit (Fr) of watermelon via quantitative real-time PCR analysis (qRT-PCR). Histochemical GUS assays of ProClHSP22.8::GUS in Arabidopsis. GUS protein was expressed in 7-day-old seedlings with GUS-PBI101 (mock) (B) and ProClHSP22.8:: GUS (C). (D–H) represents the GUS signal was expressed in root (D), stem (E), leaf (F), flower (G) and silique (H) of adult Arabidopsis plants.

Response patterns of watermelon ClHSP22.8 to ABA and salt stress

The expression patterns of ClHSP22.8 in response to ABA and salt stress were detected by qRT-PCR in watermelon leaves at four different timepoints (0, 1, 4 and 12 h). Results indicated that ClHSP22.8 expression was obviously downregulated by 0.52-and 0.14-fold at 4 and 12 h after ABA treatment, respectively (Fig. 3A). Similarly, the expression of ClHSP22.8 was significantly reduced by salt stress and reached a minimum at 4 h (0.16-fold), although the level of reduction appears to weaken at 12 h (0.61-fold) (Fig. 3B).

Figure 3 Expression profiles of ClHSP22.8 in response to abscisic acid (ABA) and salt stress.

ClHSP22.8 expression levels in watermelon leaves exposed to 100 μM ABA (A) and 200 mM NaCl (B) at 0, 1, 4 and 12 h via qRT-PCR. The asterisks on the top of the columns indicate significant differences from the value at 0 h. **P < 0.01 (C) Cis-elements analysis of ClHSP22.8 promoter sequence. (D) Histochemical GUS assays of 7-day-old transgenic seedlings with empty vector. (E and F) Histochemical GUS assays of 7-day-old transgenic seedlings with ProClHSP22.8::GUS exposed to ABA and NaCl for 24 h.

Cis-acting elements of the ClHSP22.8 promoter sequence were subsequently analyzed via PlantCARE website. Two ABA responsiveness elements (ABREs), one anaerobic responsiveness element (ARE) and two MeJA-responsiveness elements (TGACG-motifs) were identified (Fig. 3C); Two ABRE elements were located at −501 and −379 bp; the ARE element was located at −1,524 bp; and two MeJA-responsiveness elements were located at −1,477 and −381 bp. We were not able to detect an HSE motif in the promoter.

ProClHSP22.8::GUS analysis showed that there were no significant differences in terms of GUS staining in roots of 7-day-old seedlings in response to ABA and salt treatment at 24 h when compared with untreated seedlings, although there was a significant reduction in the leaves (Figs. 3D–3F).

Overexpression of ClHSP22.8 conferred ABA sensitivity to Arabidopsis

We constructed Arabidopsis lines that over-expressed ClHSP22.8 and screened these lines by both PCR and qRT-PCR (Fig. S2; Fig. 4). Compared to the WT as a control, we found that the ClHSP22.8 gene was significantly over-expressed by 8.60-, 20.32-, 16.38- and 27.20-fold in the OE22.8-1, OE22.8-2, OE22.8-3 and OE22.8-4 lines, respectively (Fig. 4A). To determine the ABA sensitivity of the lines that over-expressed ClHSP22.8, we grew the OE22.8 lines, along with the WT plants, on 1/2 MS medium with 100 μM of ABA. After 7 days, we found that the root growth of the WT plants had decreased by 29.61% in response to exogenous ABA treatment when compared with untreated WT seedlings (defined as mock). However, we found that root growth in the OE22.8-1, OE22.8-2, OE22.8-3 and OE22.8-4 seedlings was seriously repressed, by 36.88–61.08%, when compared to the WT plants (Figs. 4B–4E). This data indicated that plants that over-expressed ClHSP22.8 were more sensitive to ABA treatment.

Figure 4 Seedling assay of ClHSP22.8-overexpressing lines and wild type (WT) in response to ABA and NaCl treatment.

(A) Relative gene expression of ClHSP22.8 in different overexpression Arabidopsis transgenic lines OE22.8-1 to OE22.8-4. (B) The root length of the OE22.8 transgenic and WT plants in the presence of 100 mM ABA and 200 mM NaCl for 7 days after germination, respectively. Growth of the transgenic and WT plants under normal condition (C) and in the presence of 100 mM ABA (D) and 200 mM NaCl (E) for 7 days after germination. * and ** represent significant differences between WT and OE22.8 lines at values of P < 0.05 and P < 0.01, respectively, as determined by Student’s t-test.

Overexpression of ClHSP22.8 reduced tolerance to salt in transgenic Arabidopsis

In order to further investigate the specific roles of ClHSP22.8 in response to salt stress, we subjected 7-day-old seedlings that over-expressed ClHSP22.8 (OE22.8-1, OE22.8-2 and OE22.8-4 lines) and WT plants to 200 mM NaCl. After 7 days, the seedlings that over-expressed ClHSP22.8 showed more chlorosis and stunted phenotypes; their primary root lengths were also more significantly reduced than the WT plants (Fig. 4). Compared with mock seedlings, we found that the root length in WT plants after salt treatment was reduced by 8.87% in response to salt treatment. In contrast, the OE22.8 lines showed a more serious reduction in root length (19.35–50.43%). This data indicated that ClHSP22.8 negatively regulates plant salinity stress response.

Some ABA- and stress-related genes were repressed by over-expression of ClHSP22.8 under salt stress

Finally, we used qRT-PCR to investigate the expression profiles of several representative genes that are involved in ABA biosynthesis and signaling and stress-responsive transcription factors (TFs), including Arabidopsis 9-cis epoxycarotenoid dioxygenase 3 (AtNCED3), ABA insensitive 4 (AtABI4), ethylene response factor 05 (AtERF05), Arabidopsis dehydration-responsive element-binding protein 1B (AtDREB1B), zinc finger protein (AtZAT7) and myb domain protein 44 (AtMYB44). In the WT plants, we found that AtNCED3, AtABI4, and AtERF05 were repressed by 0.28–0.57-fold, while AtDREB1B, AtMYB44 and AtZAT7, were induced by 2.22–4.35-fold after salt treatment (Fig. S3). These results indicate that these six genes are salt-responsive genes. Furthermore, the expressions of these six genes were significantly repressed in the OE22.8-2 (from 0.47- to 0.58-fold) and OE22.8-4 (from 0.32- to 0.46-fold) lines compared to that in the WT plants (Fig. 5). This data indicated that the overexpression of ClHSP22.8 could repress the expression of these stress-responsive genes in Arabidopsis.

Figure 5 Expression profiles of ABA- and stress-related genes in WT and ClHSP22.8 overexpressing Arabidopsis plants.

(A–F) Represent the expression profiles of Arabidopsis 9-cis epoxycarotenoid dioxygenase 3 (AtNCED3), ABA insensitive 4 (AtABI4), Arabidopsis dehydration-responsive element-binding protein 1B (AtDREB1B), ethylene response factor 05 (AtERF05), myb domain protein 44 (AtMYB44), and zinc finger protein (AtZAT7) respectively. * and ** represent significant differences from the control at values of P < 0.05 and P < 0.01, respectively, as determined by Student’s t-test.

In order to confirm whether the reduction/elevation in expression was more pronounced in the OE22.8 lines when treated with 200 mM NaCl, we normalized the changes observed in WT plants and then checked if the OE22.8 lines showed statistically significant differences in terms of gene expression (Fig. 6). We observed a significant reduction in the expression of AtNCED3, AtABI4 and AtDREB1B in the OE22.8-2 and OE22.8-4 lines after salt treatment. The expression levels of AtERF05 and AtMYB44 were obviously repressed, but only in the OE22.8-4 line. There were no significant differences in the expression levels of AtZAT7 in the OE22.8 lines after salt treatment. These results indicate that all of the detected genes, except AtZAT7, might represent target genes for ClHSP22.8-mediated repression in response to salt stress.

Figure 6 Expression profiles of ABA- and stress-related genes in three-week-old WT and ClHSP22.8 overexpressing Arabidopsis plants 24 h after salt treatment.

(A–F) Represent the expression profiles of AtNCED3, AtABI4, AtDREB1B, AtERF05, AtMYB44, and AtZAT7 respectively. The changes observed in WT were normalized. * and ** represent significant differences between the mock- and salt-treatment OE22.8 plants at values of P < 0.05 and P < 0.01, respectively, as determined by Student’s t-test.

Discussion

Salinity is an important environmental stress factor and can have a severe effect on the growth and development of plants. Consequently, salt stress is a growing problem for global agricultural production (Abbasi et al., 2016). Watermelon (Citrullus lanatus) is a salt-sensitive crop and may help us to engineer more salt-tolerant varieties so that we can investigate the core salt-tolerance mechanisms in this fruit (Yetr & Uygur, 2009). As the largest family, and most well studied HSP, the HSP20 family of proteins is ATP-independent and generally assemble into large oligomers that can protect other proteins from denaturation and aggregation (Papsdorf & Richter, 2014; Asea, Kaur & Calderwood, 2016). An increasing body of evidence has shown that HSP20s regulate the responses of plants to environmental changes and thus allow plants to survive adverse conditions (Sun et al., 2012, 2016, 2020; Murakami et al., 2004; Charng et al., 2006; Shakeel, Heckathorn & Luthe, 2012; Khurana, Chauhan & Khurana, 2013; Zou et al., 2012; Kaur et al., 2015; Chauhan et al., 2012; Wang et al., 2020; Mu et al., 2013; Li et al., 2018). In a previous study, we identified HSP20 genes within the genome of watermelon and found that HSP20s in plants can be divided into 18 subfamilies (He et al., 2019). The largest group is the nucleocytoplasmic (C)-located HSP20s; this group features 13 subfamilies that exhibit functional redundancy and divergence (He et al., 2019; Guo et al., 2020; Kim et al., 2011). As a member of the CIX subfamily, ClHSP22.8 has a ACD domain, agglomerates into granules in the cytoplasm, and existes as larger oligomers in vivo as expected (He et al., 2019). Gene duplication events are major sources of new gene functions (Francino, 2005). ClHSP22.8 was duplicated with ClHSP17.6C, ClHSP17.6D, and ClHSP16, from the CI subfamily, but it does not have a close relationship with the duplicates, or homologs in other species, except for CsHSP23.7 in cucumber (Fig. 1B; Table S1) (He et al., 2019). Notably, we found ClHSP22.8 could not be induced by heat but was significantly repressed by salt stress (Fig. 3B), which was different from a typical HSP20 that exhibited rapid and significant upregulation under heat stress (Papsdorf & Richter, 2014; Asea, Kaur & Calderwood, 2016; He et al., 2019). Therefore, we consider that ClHSP22.8 probably evolved new functionality in stress responses following the gene duplication event (Francino, 2005).

In order to verify the precise functional role of ClHSP22.8 in salt stress response, we created lines of Arabidopsis that over-expressed ClHSP22.8 and found that these overexpression lines exhibited shorter roots and more yellow leaves under salt stress. Thus, these results indicate that ClHSP22.8 negatively regulates the salinity tolerance of Arabidopsis (Fig. 4). In this study, we used qRT-PCR and promoter::GUS analysis analysis and found that ClHSP22.8 was clearly repressed by salt treatment (Fig. 3B). The transcript abundance of ClHSP22.8 in response to salt stress reached the lowest level in the first 4 h. The response pattern of ClHSP22.8 that rapidly and sharply responded to salt stress in a short time and then had slight variations, was similar to quite a few HSP20s such as TaHSP23.9 (Wang et al., 2020), OsHSP20 (Guo et al., 2020), AsHSP17 (Sun et al., 2016) and so on. The response pattern of HSP20s probably is a kind of mechanism for plants rapidly adapt to salt stress.

HSP20s are known to modulate the multiple signaling pathway so as to regulate the plant’s response to salt stress. Most HSP20s have been reported to play positive roles in regulating plant tolerance to salt, including maize ZmHsp16.9 (Sun et al., 2012), rice OsHSP17.0 and OsHSP23.7 (Zou et al., 2012), wheat TaHSP23.9 (Wang et al., 2020), David Lily LimHSP16.45 (Mu et al., 2013) and sweet pepper CaHsp22.5 (Li et al., 2018). However, recent work, a few studies about the negative effect of HSP20s on plant response to salt stress have been reported. The over-expression of AsHSP17 or AsHSP26.8 in plants led to the direct repression of the vast majority of stress-responsive genes involved in plant photosynthesis, ABA-dependent and ABA-independent pathways, and some other stress response pathways, thus led to reduced levels of salt tolerance (Sun et al., 2016, 2020), but the negative regulation mechanisms of HSP20s in salt stress responses remain to be fully unraveled. Similar to AsHSP17 and AsHSP26.8 (Sun et al., 2016, 2020), ClHSP22.8 also plays negative roles in terms of salt response involved in ABA signaling pathway. ABA responsiveness element (ABRE) and anaerobic response element (ARE) cis-elements can be recognized by AREB/ABF and MYB transcription factors, respectively (Banerjee & Roychoudhury, 2017; Fujita et al., 2011). The two cis-elements are necessary for ABA- and anaerobic-responsive gene expression (Banerjee & Roychoudhury, 2017; Fujita et al., 2011). In the present study, two ABRE and one ARE elements were identified in the ClHSP22.8 promoter (Fig. 3C). And ClHSP22.8 was significantly repressed by the exogenous ABA via qRT-PCR and promoter::GUS analysis (Figs. 3A, 3D and 3E). Above results indicate that ClHSP22.8 respond to salinity stress in a negative manner and that this response involved the ABA mediated signaling pathway.

To further illustrate the regulatory mechanism of ClHSP22.8 under salt stress, we used qRT-PCR to determine the expression patterns of six ABA- and stress-related genes in response to salt stress; results demonstrated that all of these detected genes could obviously respond to salt stress which AtNCED3, AtABI4, and AtERF05 were obviously repressed while AtDREB1B, AtMYB44, and AtZAT7 were induced by salt stress (Fig. S3). Meanwhile, studies found that over 90% differentially expressed genes (DEGs) in AsHSP17 and AsHSP26.8 overexpressed lines were down-regulated (Sun et al., 2016, 2020). Similarly, all of the detected genes in this study were significantly repressed by overexpression of ClHSP22.8. These results indicate all of the six detected genes are salt-responsive genes and repressed by ClHSP22.8.

ABA-dependent pathways is an important mechanism in adaptation to salt stress and affacts plant salt stress response and tolerance (Roychoudhury, Paul & Basu, 2013; Yoshida, Mogami & Yamaguchi-Shinozaki, 2014). ABA biosynthesis and signalling and some stress-responsive transcription factors involved in ABA signalling were key regulators in ABA-dependent pathway. Among the detected genes in this study, AtNCED3 encodes a rate-limiting enzyme that plays a role in ABA biosynthesis and ABI4 is a key regulator of the ABA-dependent pathway (Roychoudhury, Paul & Basu, 2013; Yoshida, Mogami & Yamaguchi-Shinozaki, 2014). Their expression were significantly repressed in OE22.8 lines after normalized the changes observed in WT plants (Fig. 6). So the over-expression of ClHSP22.8 probably resulted in lower levels of ABA and enhanced ABA sensitivity by repressing the expression of AtNCED3 and ABI4 transcripts in response to salt stress. Besides, some MYB and ZAT transcription factors have been implicated in the plant response to abiotic stress and ABA sensitivity (Ciftci-Yilmaz et al., 2007; Nguyen & Cheong, 2018; Yu et al., 2017; Wei et al., 2017; Seo et al., 2012; Persak & Pitzschke, 2014). AtMYB44 was involved in ABA-dependent signaling pathways that regulate stress adaption and confer plant tolerance to salt stress (Nguyen & Cheong, 2018; Seo et al., 2012; Persak & Pitzschke, 2014). And the constitutive expression of AtZAT7 suppressed growth and enhanced salt tolerance in transgenic Arabidopsis plants (Ciftci-Yilmaz et al., 2007). AtMYB44 and AtZAT7 have both been shown to be regulated by HSP20s in response to salt stress (Sun et al., 2016, 2020). In this study, we found that the expression of AtMYB44 and AtZAT7 can be obviously induced by salt in WT plants but the reduction of AtMYB44 is more pronounced in the OE22.8-4 plants (Fig. S3; Fig. 6). These results indicated that ClHSP22.8 participates in the response to salt stress via an ABA-dependent pathway.

Our data suggested ClHSP22.8 also modulated the response to salt stress via an ABA-independent signaling pathway. Previous work has shown that AtDERB1B and AtERF05 participate in ABA-independent pathway (Sun et al., 2020; Yoshida, Mogami & Yamaguchi-Shinozaki, 2014); and we found AtDERB1B and AtERF05 were both ClHSP22.8-regulated salt responsive genes. The expression of AtDERB1B were obviously induced while AtERF05 were repressed by salt stress in WT plants (Fig. S3). And they were both significantly repressed in OE22.8 lines (Fig. 5). In further, the two genes were significantly repressed by salt stress in OE22.8 lines after normalizing the effects shown in WT plants (Fig. 6).

Collectively, these results imply that, as a negative regulator of salt stress, ClHSP22.8 may be repressed to an appropriate level in protecting plants from salt stress. However, when ClHSP22.8 was overexpressed in Arabidopsis, some genes involved in ABA-dependent (AtNCED3 and ABI4) and ABA-independent (AtDERB1B and AtERF05) signaling pathways, and stress-responsive TF (AtMYB44) in ABA signalling were repressed, and then the salt stress response regulatory network was negatively impacted as outlined in Fig. 7. Our data provides further understanding of the specific roles of HSP20s in the watermelon in terms of the response to abiotic stress.

Figure 7 A proposed model for the roles of ClHSP22.8 in salt stress resistance in Arabidopsis.

ClHSP22.8 is involved in ABA biosynthesis, ABA-dependent and independent, and other stress responsive signaling pathways to modulate plant response to salt stress. Abbreviations: AtNCED3, Arabidopsis 9-cis epoxycarotenoid dioxygenase 3; AtABI4, Arabidopsis ABA insensitive 4; AtERF05, ethylene response factor 05; AtDREB1B, Arabidopsis dehydration-responsive element-binding protein 1B; AtMYB44, Arabidopsis myb domain protein 44. Arrows indicate positive regulation, whereas lines ending with a bar indicate negative regulation.

Conclusions

In summary, we identified an HSP20 gene, ClHSP22.8, and demonstrated that this gene plays important roles in the salt response. Analyses involving both qRT-PCR and promoter::GUS analysis indicated that the expression of ClHSP22.8 in Arabidopsis could be repressed by exogenous ABA and salt stress. Furthermore, the over-expression of ClHSP22.8 repressed some genes that are known to be involved in ABA-dependent and independent signaling pathways, and other stress-responsive pathways, thus leading to an enhanced level of plant sensitivity to ABA and a reduced tolerance to salt stress. Our study provides a better understanding of the specific roles of HSP20s in watermelon with regards to the abiotic stress response and suggests that ClHSP22.8 may be a valuable gene to consider when cultivating watermelons.

Supplemental Information

Supplemental Information 1 Schematic diagram of the ClHSP22.8 gene expression construct.

(A) p35S:: ClHSP22.8/p35S-KanR. The ClHSP22.8 gene (coding sequence only), and a resistance gene, kanamycin (KanR), were both under the control of the CaMV35S promoter. RB, right border; LB, left border (B) Schematic diagram of the ProClHSP22.8::GUS-PBI101 vector.

Click here for additional data file.

Supplemental Information 2 PCR analysis of ClHSP22.8 in Arabidopsis wild type (WT) and OE22.8 transgenic lines.

Lanes 1 to 7 indicate the DNA marker, WT, p35S::ClHSP22.8 vector, and OE22.8-1 to OE22.8-4, respectively.

Click here for additional data file.

Supplemental Information 3 Expression patterns of ABA- and stress-related genes in response salt stress via quantitative real-time PCR (qRT-PCR).

The expression patterns of Arabidopsis 9-cis epoxycarotenoid dioxygenase 3 (AtNCED3), ABA insensitive 4 (ABI4), ethylene response factor 05 (AtERF05), Arabidopsis dehydration-responsive element-binding protein 1B (AtDREB1B), zinc finger protein (AtZAT7), and myb domain protein 44 (AtMYB44) were analyzed. * and ** represent significant differences from the control at values of P < 0.05 and P < 0.01, respectively, as determined by Student’s t-test.

Click here for additional data file.

Supplemental Information 4 Primers used in this study.

Click here for additional data file.

Supplemental Information 5 The basic information of ClHSP22.8 in watermelon.

Click here for additional data file.

Supplemental Information 6 Raw data used in qRT-PCR analyses and Figs. 2A, 3A, 3B, 4A, 5 and 6.

Click here for additional data file.

Additional Information and Declarations

Competing Interests

Author Contributions

Data Availability

The authors declare that they have no competing interests.

Yanjun He conceived and designed the experiments, performed the experiments, analyzed the data, prepared figures and/or tables, and approved the final draft.

Yixiu Yao performed the experiments, prepared figures and/or tables, authored or reviewed drafts of the paper, and approved the final draft.

Lili Li performed the experiments, analyzed the data, authored or reviewed drafts of the paper, and approved the final draft.

Yulin Li performed the experiments, prepared figures and/or tables, and approved the final draft.

Jie Gao analyzed the data, authored or reviewed drafts of the paper, and approved the final draft.

Min Fan conceived and designed the experiments, analyzed the data, authored or reviewed drafts of the paper, and approved the final draft.

The following information was supplied regarding data availability:

The raw data are available as Supplemental Files.

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
