# Peer review of "A heat-shock 20 protein isolated from watermelon (ClHSP22.8) negatively regulates the response of Arabidopsis to salt stress via multiple signaling pathways"

_PeerJ, doi:10.7717/peerj.10524_

## Round 0.1 · original submission · Major Revisions

Dear authors,

I received comments on your paper from experts in the field. Below are their suggestions to improve the scientific quality of your manuscript.
Please, I invite you to detail as much as possible the statistical section, which is in the present form very poor. The results need to be also detailed and the discussion section needs further work and comparisons to the large literature. I invite you to carefully consider these important points.

Kind regards
Dr. Gagaoua M

Reviewer 1 ·

Basic reporting

Aims and objectives are clearly defined. Clear outline of research. References are relevant and properly cited.

Experimental design

Clear description of methods. Sound experimental design.

Validity of the findings

The findings reported here have a potency to open new research avenues in the field.

Additional comments

The manuscript submitted by He et al., presents a study focusing on gene expression regulation of HSP20 – ClHSP22.8 from watermelon. The authors show that the expression of this gene is repressed under stress conditions like exogenous abscisic acid (ABA) and salt stress and it acts as a negative regulator of plant stress response.
Although the overall data are interesting, there are a few limitations that need to be addressed. Here are my comments that might help authors to improve the manuscript quality.

1) Clear and convincing data for the role of ClHSP22.8 in ‘stress induced repression/activation’ of target genes are missing. Figure 5 and lines 206-224 describe the effects on the levels of ABA- and stress-related genes under salt treatment upon overexpression of ClHSP22.8. These claims need to be confirmed by normalizing the effects seen in WT cells. All the changes in the levels of the various genes tested, are observed in both WT and OE22.8 cell lines. In order to confirm the reduction/elevation is more pronounced in the OE22.8 lines, authors should normalize the changes observed in WT cells and then check if OE22.8 lines show statistically significant difference in regulation.
2) Figure 5: Did authors employ the rest of the two OE22.8 lines? Are these results reproducible?
3) Lines 179:182: Authors describe that there are two ABA responsiveness elements (ABREs) and one anaerobic induction element (ARE) from the ClHSP22.8 promoter (Figure 3C). The biochemical validation of the significance of these elements is necessary.
4) Did authors employ any negative control or empty vector in the experiments described in Figure 2B?
5) Figure 4: Growth of the transgenic and WT plants under ’mock’ treatment is an important control and needs to be shown like in Figures 4A and 4B.
6) Authors need to comment on the result where there is around 4-5 fold induction in the levels of ClHSP22.8 expression after 12 hours of NaCl treatment.
7) Figure 4B: Authors claim to observe a reduction in root growth in WT plants by 29.61% and 8.87% by ABA and salt treatment respectively. Are these reduction values statistically significant?
8) Figure S2: Is this image acquired from the same gel? The lane 1 and the other lanes seem to be spliced together. Also, what was the loading and/or normalization control used here?
9) Figure 4A: The data seem to be normalized to OE22.8-1 expression, is that correct? In my opinion, they should be normalized to WT levels.
10) Figure 4B: The data for OE22.8-3 are missing.
11) Discussion includes many details regarding the implications of the findings and speculates possible mechanisms. Can authors include a schematic model?
12) Figures 3A, 3B: The Y axis label is missing. Also, an asterisk (*) representing statistical significance is missing.
13) The legends for supplementary figures are missing.
14) The font size of the data labels in many figures is too small.
15) The quality of some the figures is poor; so high resolution images are needed.
16) A grammar check would be helpful.

Reviewer 2 ·

Basic reporting

This manuscript is well-prepared and nicely presented with good scientific quality. However, below I have highlighted some issues, which need to be fixed before further consideration. Also, see the attached file for some minor corrections.

Experimental design

See the below comments.

Validity of the findings

See the below comments.

Additional comments

 Correct the spelling of multiple in the title.
 The word Arabidopsis should always be in italic in the title and main text body. Please check the entire text and fix it.
 Add a space before the citation. There are many places where spaces need to be inserted. Please carefully check the entire text for this error and fix it.
 Line 56, 57, define the ABA on the first appearance.
 Line 60-65, we need strong references for these statements. Please check and add a suitable reference.
 Line 121, 136, use the standard style for units, i.e., 100 mg L-1. Check the entire text/figures and modify the unit.
 Line 153-160, please rearrange the text and cite the figure 1 accordingly. Currently, Fig 1B is not cited in the text; probably authors forget to write 1B.

Annotated reviews are not available for download in order to protect the identity of reviewers who chose to remain anonymous.

Reviewer 3 ·

Basic reporting

The article is interesting. It analyzed the expression patterns and roles of ClHSP22.8 from watermelon in response to salt stress. The article found that ClHSP22.8 is a negative regulator in plant response to salt stress via ABA-dependent and -independent, and other stress responsive signaling pathways. But it still needs some editing.

Experimental design

Generally I think the experimental design is proper.

Validity of the findings

no comment

Additional comments

The manuscript (MS) by He et al. titled " A heat-shock 20 protein isolated from watermelon (ClHSP22.8) negatively regulates the response of Arabidopsis to salt stress via multiple signaling pathways " characterized an HSP20 gene (ClHSP22.8) from watermelon. Quantitative real-time PCR (qRT-PCR) and promoter::β-glucuronidase (GUS) assays showed that the expression of ClHSP22.8 was repressed by exogenous ABA and salt treatment. Meanwhile, Arabidopsis lines that over-expressed ClHSP22.8 were successfully constructed and used to investigate the sensitivity to ABA and salt tolerance. The expression profiles of genes related to ABA and the stress response of plants that over-expressed ClHSP22.8 under salt treatment were analyzed. The MS is well written, which not only provide a better understanding of the specific roles of HSP20s in watermelon adaption to salt stress, but may also provide insight into the potential signaling processes in response to stressful conditions. There are several minor issues that should be corrected:
1. In the title ” A heat-shock 20 protein isolated from watermelon (ClHSP22.8) negatively regulates the response of Arabidopsis to salt stress via mutiple signaling pathways ” ‘mutiple’ should be ‘multiple’.
2. In the Figure 3A, the legend of At, Os… was absent.
3. In the Figure 4, the name of the over-expressed ClHSP22.8 lines (OE22.8-1, OE22.8-2, and OE22.8-4) should be uniform.

---

## Round 0.2 · accepted · Accept

Dear authors,

I am glad to inform you that your paper is now accepted for publication in PeerJ.

Kind regards
Dr. Gagaoua M

Reviewer 1 ·

Basic reporting

The manuscript meets basic criteria of reporting.

Experimental design

Sound experimental design has been employed. I do not have any additional comments on the revised version of the manuscript.

Validity of the findings

No additional comments on the revised manuscript.

Additional comments

The revised version of the manuscript submitted by He, Yao et al., addressed the reviewers' concerns raised in the earlier version of the manuscript. I recommend this article for publication in PeerJ.

Reviewer 2 ·

Basic reporting

Dear Authors,
Thank you for revising the MS according to the proposed comments and suggestions. Notably, the MS has been improved; thus, I am endorsing the submission for publication in PeerJ. Congratulations!

Experimental design

no comment

Validity of the findings

no comment

Additional comments

no comment